POINT OF VIEW

# Unbridle biomedical research from the laboratory cage

**Abstract** Many biomedical research studies use captive animals to model human health and disease. However, a surprising number of studies show that the biological systems of animals living in standard laboratory housing are abnormal. To make animal studies more relevant to human health, research animals should live in the wild or be able to roam free in captive environments that offer a natural range of both positive and negative experiences. Recent technological advances now allow us to study freely roaming animals and we should make use of them.

**GARET P LAHVIS**[*]

**\*For correspondence:** lahvisg@ohsu.edu

For over 50 years, biologists have asked how the addition of a few objects to a laboratory cage can change the biology of a laboratory rodent. We have learned that even modest amendments to cages lead to vast changes in the neurobiology, behavior, immune responses, disease resistance and cancer remission of captive mice and rats. This so-called environmental 'enrichment' makes research animals more resilient and helps them to recover from many kinds of experimental treatments. These studies often conclude that enrichment serves as a form of therapy, but they overlook an important point. The cage floor areas provided to rodents and primates used in research are much smaller than their natural ranges. The captive environments of research animals also deny them ongoing opportunities to explore and learn from the variety of rewards and challenges that they would normally experience in the wild.

By studying organisms confined to restricted environments that are unresponsive to their actions, biomedical research violates one of its core propositions – that animals used as experimental controls embody healthy biological systems. We now have the technologies to study free-roaming organisms in the wild or in captivity under naturalistic conditions. Advances in remote recording and transmission of data (telemetry) allow us to transfer data wirelessly so we can modulate the physiology of free-roaming animals and record their responses. By using these technologies, we can provide research animals with a greater sense of wellbeing and make animal studies more relevant to human health and disease.

## Captivity alters animal development

In the early 1960s, Mark Rosenzweig and his colleagues asked how environmental changes affected the brains of his laboratory rats. Each day, they placed new wooden blocks inside the rats' cages and let the rats explore mazes that were reconfigured every day. The team discovered that the sensory cortex, a region of the brain that processes all of the senses (sight, sound, touch, smell and taste), was larger in these experimental rats than in the rats living in standard laboratory cages (*Rosenzweig et al., 1962*).

Since then, researchers have learned that even modest changes to the conditions in a standard laboratory cage – such as the one-time addition of objects, enhanced maternal care, or more frequent handling – lead to changes in the animals' brains. This enrichment alters the densities and appearance of neurons (*Kempermann et al., 1997*; *Greenough et al., 1973*), other brain cells (*Szeligo and Leblond, 1977*; *Viola et al., 2009*) and blood vessels (*Black et al., 1987*). Strikingly, enrichment promotes changes in some regions of the brain, but not others (*Szeligo and Leblond, 1977*;

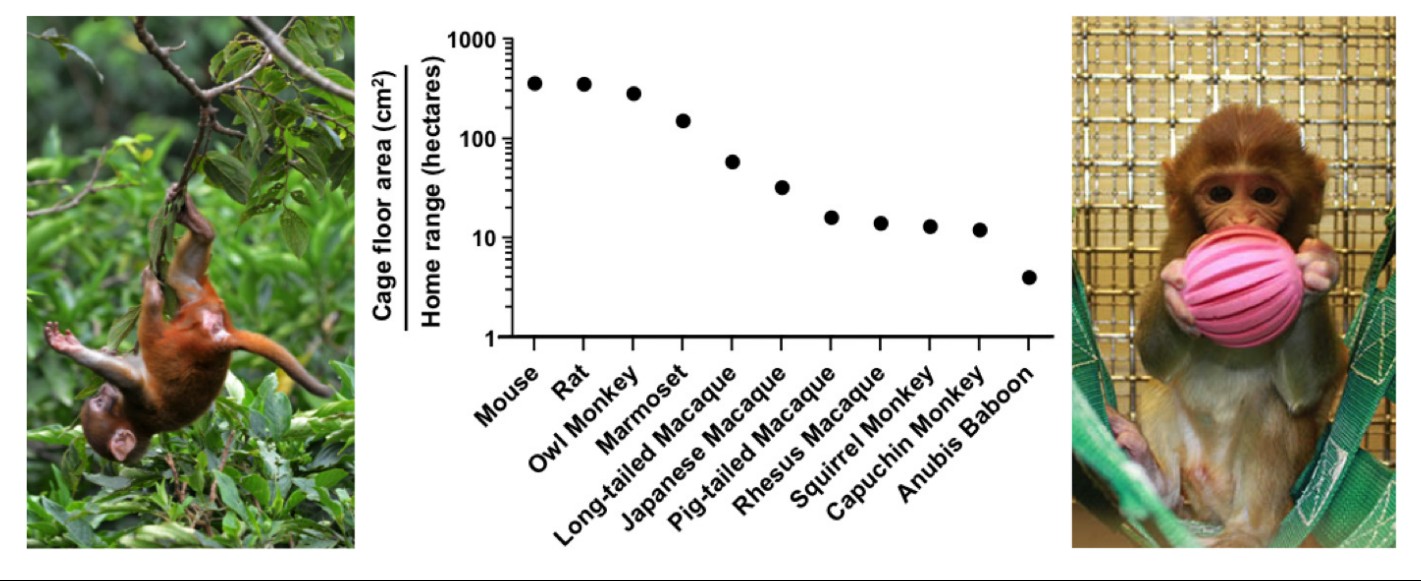

**Figure 1.** Laboratory cages are much smaller than an animal's natural home range. The middle panel shows the ratio of the recommended cage floor area (cm$^2$; *Albus, 2012*) to the estimated natural home range in hectares (10,000 m$^2$; *Nunn and Barton, 2000*; *Chambers et al., 2000*; *Bramley, 2014*) for eleven species used in research. The left panel shows a juvenile rhesus macaque in a Nepali jungle. The right panel shows a juvenile rhesus macaque in an enriched cage.
IMAGE CREDIT: photo of the caged juvenile macaque courtesy of the University of Wisconsin Primate Center.

*Faherty et al., 2003*). This structural reorganization is mediated by local differences in gene expression (*Pham et al., 1999*) that can persist throughout life (*Thiriet et al., 2008*; *Meaney, 2010*; *Mychasiuk et al., 2012*).

Laboratory animals are less sensitive to experimental procedures when they have something to do. For example, enrichment can help animals to recover from brain trauma and seizures (*Passineau et al., 2001*; *Koh et al., 2007*), reverse brain damage caused by exposure to chemicals (*Goldberg et al., 2011*; *Guilarte et al., 2003*; *Shih et al., 2012*), and make rodents less susceptible to the effects of recreational drugs (*Stairs and Bardo, 2009*; *Solinas et al., 2010*). These studies have prompted most researchers to conclude that enrichment serves as a form of therapy (*Lahvis, 2016a*).

Critically, the term enrichment implies that the conditions imposed by standard laboratory housing are somehow normal. Laboratory cages offer poor environments relative to the natural conditions where our research animals evolved. For example, the floor area provided to a mouse inside a standard cage is 280,000 fold smaller than its natural home range. This difference becomes even more extreme for larger animals used in research such as rhesus macaques (7

million fold) and Anubis baboons (25 million fold; *Figure 1*).

An essential role of the brain is to adjust behavior to an unpredictable environment, an everyday experience common to wild animals and humans. By contrast, even enriched cage environments offer none of the temporal variations occurring in nature, such as changes in temperature and humidity, the availability of food and shelter, and risks to survival. These comforts and adversities evoke a wealth of feelings, including pain, fear, hunger, anticipation and pleasure. The caged environment, whether standard or enriched, isolates the research animal from these everyday experiences.

We know that depriving research animals of physical experiences and visual stimuli decreases their respective activities in the motor and visual cortex (*Ostrovsky et al., 2006*; *Strata et al., 2004*). Likewise, it is untenable to assume that a brain deprived of a complex environment of rewards and challenges represents that of a healthy animal or human.

## Responding to a responsive environment

By contrast to what happens in a laboratory cage, wild animals behave in ways that aim to

optimize energy uptake or minimize the time they spend foraging (*Krebs and Davies, 2009*). An animal must consume more energy than it expends (*Fonseca et al., 2014*; *Belovsky, 1984*) and the balance of this relationship is sensitive to other factors, such as risks of predation (*Hoogland, 1981*), requirements for nutrients (*Belovsky, 1978*) or the temporary need for medicinal plants (*McPherson, 2013*). Optimal foraging models account for an animal's moment-to-moment decisions of what paths to take, what foods to eat and how much time to spend foraging versus watching for predators.

Humans and other animals make decisions based on our 'affective' experiences: we seek rewards and avoid situations that we feel are unpleasant or potentially harmful (*Schneirla, 1959*). Within a complex and unpredictable natural environment, an organism learns what cues and contexts are associated day-to-day with rewarding or unpleasant situations. These affective experiences inform and respond to ongoing decisions.

Consider, for example, the temporary and patchy distributions of food, such as fruits ripening and decaying, that are thought to have favored the evolution of primate intelligence (*Navarrete et al., 2016*). A fruit-eating primate must make a cognitive decision, choosing a course of action from various alternatives in the context of its affective experience, such as the desire to taste a particular kind of fruit. In nature, a primate might abstain from eating local food items to follow spatial cues to more nutritious or more flavorful foods at distant locations. As these distant foods decay or become depleted, the primate must then learn that the spatial cues are no longer predictive. Even under simpler foraging conditions, individual animals of many species must learn and relearn which spatial cues predict the locations of food items that are only available in particular places for short periods of time.

Wild animals also consider various factors in decisions not directly related to foraging. These include habitat selection (*Raynor et al., 2017*), mate selection (*Candolin, 2003*), cooperation with other individuals (*Lahvis, 2016b*), and predators (*Lagos et al., 2009*). Such challenges are common for wild animals (*Krebs and Davies, 2009*) and resemble the everyday conflicts of human experience. By contrast, laboratory animals are denied opportunities to make decisions within a natural context that offers a breadth of spatial and temporal variety.

Some argue that laboratory mammals, particularly research primates living in open colonies, have opportunities to navigate complex social interactions inside their corrals and thus experience challenges that are akin to those of their wild counterparts. However, conservation biologists find that captive-raised animals reintroduced into the wild – such as California condors, spotted owls and Pacific salmon – fail to express the cognition, motivation and behaviors necessary to survive and reproduce (*Reading et al., 2013*).

Some argue we need to adhere almost exclusively to studies of caged animals to control irrelevant environmental variables that, if left unchecked, complicate the delicate cause-effect relationships governing development. Yet, we often lack or fail to control many of the most important environmental variables inside laboratory cages, such as food and bedding. Standard housing fosters increased sensitivity to stimuli that researchers have not noticed or find difficult to control. For example, variations in animal feed (*Garey et al., 2001*; *Brown and Setchell, 2001*), bedding (*Burn et al., 2006*; *Robinson et al., 2004*), the sex of the experimenter (*Bohlen et al., 2014*; *Sorge et al., 2014*), or ultrasonic noises (*Turner et al., 2005*; *Milligan et al., 1993*) can render behavioral tests difficult to replicate. Indeed, mouse behaviors are difficult to reproduce across laboratories in standard cages (*Crabbe et al., 1999*; *Richter et al., 2010*) and are more reproducible when housing conditions are varied (*Würbel, 2002*; *Richter et al., 2009*, *2010*).

## Not just neurobiology

Relative to laboratory animals housed in enriched cages, animals housed in standard cages express greater levels of cortisone, a hormone that affects immune cells and impairs immune responses throughout the body. These changes reduce the ability of a captive animal to mount an effective response to infections (*Arranz et al., 2010*; *Gurfein et al., 2014*). Conventional caging also promotes obesity, type II diabetes and high blood pressure (*Martin et al., 2010*), while decreasing muscle strength and endurance (*Sirevaag and Greenough, 1987*; *Spangenberg et al., 2005*; *During et al., 2015*). Furthermore, standard laboratory housing enhances tumor growth in animal models of several cancers (*Cao et al., 2010*; *Li et al., 2015*; *Nachat-Kappes et al., 2012*). The 'control' animals used in such experiments do not represent

what biomedical research requires for its premise; they are not healthy individuals.

Few studies have compared the biology of wild versus cage-reared animals. Among caged prairie voles, variations of a gene encoding the receptor for a hormone called vasopressin are highly correlated with sexual fidelity (*Okhovat et al., 2015*). However, these correlations are not nearly as obvious among voles living in the wild. A different study shows that the structure of the visual cortex in caged Norway rats differs remarkably from that of their free-roaming counterparts (*Campi et al., 2011*). Wild animals also express lower levels of cholesterol than captive animals (*Schmidt et al., 2006*; *Eades et al., 1963*). Recently, a subset of immune cells called T cells that are expressed by humans but not found in laboratory mice, was identified in feral mice. The report stated that feral mice have "immune systems closer to those of adult humans" (*Beura et al., 2016*).

## Research into animal wellbeing

Wellbeing assumes a freedom to engage actively and proactively with responsive surroundings, exploring, problem-solving, and learning to deal skillfully and flexibly with new and existing challenges (*Špinka and Wemelsfelder, 2011*). Animal welfare scientists assert that non-human animals are intrinsically motivated to be 'doers' or 'authors' of their own activities. They argue that animals can be motivated to sample their environments, exposing themselves to nominal risks and engaging with obstacles to be overcome (*Špinka and Wemelsfelder, 2011*).

In support of these ideas, numerous studies suggest that laboratory animals prefer complex environments to predictable housing conditions. For example, animals prefer to spend more time in enriched environments than in standard environments (*Bayne et al., 1992*; *Bevins et al., 2002*; *Schroeder et al., 2014*). Rats press a lever more frequently for food rewards allocated at unpredictable levels than for regular aliquots (*Anselme et al., 2013*).

Classical conditioning approaches, used to determine whether an animal finds a stimulus rewarding or unpleasant (*Bardo and Bevins, 2000*; *Pellman and Kim, 2016*), show that laboratory rats prefer an environment where they have experienced repeated access to new objects over an environment previously paired with familiar objects (*Bevins et al., 2002*). These preferences are driven by the animal's affective experiences and, in this case, interest or enjoyment felt inside the environment paired with the new object. These affective experiences are controlled by well-defined reward circuits in the brain (*Bevins et al., 2002*; *Panksepp, 1998*).

Subjective experiences are known only to the individuals that have them (*Dawkins, 2015*). Language can only give us inferences to the subjective experiences of humans. For instance, if we both use the word 'red' to describe what we see in a ripe apple, our shared use of the same word suggests a common experience even if we both see a different color when we look at the same ripe apple (*Russell, 1912*). For humans and other animals, we can sometimes infer subjective experiences from variations in gait, gesture, facial expression and vocal intonation (*Darwin, 1872*; *Panksepp, 1998*). We can also make inferences from measures of physiological responses in the brain (e.g. *Kelley, 2005*), or of systemic responses, such as changes in the level of a stress hormone called cortisol in the body (*Detillion et al., 2004*). When we use a variety of approaches, we can infer that cages have adverse influences on the subjective experiences of captive animals (*Boissy et al., 2007*; *Yeates and Main, 2008*).

We can also have confidence that subjective experiences of humans and other animals lie on a continuum (*De Waal, 2016*; *Panksepp, 1998*). For humans, captivity inside a cage is punishment. Just as imprisonment deserves ethical consideration, we have an ethical responsibility to improve the conditions of captivity that we impose on research animals (*Gruen, 2014*). If these animals were afforded natural levels of environmental complexity, an expectation is that they would rarely attend to the outside boundaries of their cages.

Within a monotonous and predictable captive environment, where confinement is unresponsive to action, laboratory animals are deprived of their natural agency. Instead, the animals are left with boredom and, with time, learned helplessness and depression (*Špinka and Wemelsfelder, 2011*).

## What can we do?

We should replace our study of caged animals with studies of animals living in the wild or under captive but naturalistic conditions. Recent advances in technology allow us to bring traditional biomedical research to the study of freely roaming animals. For many years, wild organisms have been employed as sentinels of the

biological effects of unanticipated 'real-world' exposures to complex pollutant mixtures (*Fox, 2001*; *Basu et al., 2007*; *Ramalhinho et al., 2012*). With new technologies, we can also learn how genes and gene-by-environment interactions contribute to individual differences in disease susceptibility and resistance.

It is now possible to track free-roaming animals across large areas. We can record their behaviors with low-cost camera traps (*O'Connell et al., 2010*), lightweight radio frequency identification transponders (*König et al., 2015*) and integrated applications of accelerometers, magnetometers, and pressure sensors (*Sommer et al., 2016*). Data can be transmitted by WiFi (*Su et al., 2015*), GPS (*Jawalkar et al., 2017*), and ultra-low power sensors (*Dressler et al., 2016*). We can record the electrical activity of the brain in freely moving animals, such as the mid-flight sleep patterns of frigate birds (*Rattenborg et al., 2016*) or the activities of neurons in wild unrestrained rats (*Szuts et al., 2011*).

Miniaturized optics are available to conduct real-time, optical imaging (*Yu et al., 2015*) and optogenetics can remotely alter neuron activity (*Tye and Deisseroth, 2012*). We can use optofluidic neural probes to remotely modulate gene expression and deliver viruses, peptides, and drugs to non-human animals (*Jeong et al., 2015*). It is also possible to remotely monitor heart rate and the concentration of various molecules in tissue (*Bazzu et al., 2009*; *Rutherford et al., 2007*). Furthermore, advanced statistical approaches can be used to identify genetic and environmental factors that contribute to individual differences in biology and behavior (*de Boer et al., 2017*; *Nussey et al., 2007*).

Other advances include developing smaller sensors that are less invasive when implanted into animals and applying non-invasive imaging tools to research on small animals (*Jang, 2013*). These advances, along with the proposed improvements to environmental conditions, might be sufficient to improve the wellbeing of research animals and also provide relevant insight to a range of biological processes.

Some experiments would be difficult to conduct under wild conditions, such as manipulations that cause illness. In these cases, enclosed naturalistic environments would be necessary to give researchers the ability to rapidly identify and retrieve sick animals. Captivity would also be necessary for gene-targeted animal models that, if released, may pose a threat to the environment.

In the figure, calculations of the collapse of space available to a laboratory rat or mouse were based upon estimates of their natural home ranges (*Figure 1*). However, feral mice and rats often live adjacent to humans where food resources are concentrated. For mice, a 'naturalistic' captive environment might entail simulated barn-like arenas. Studies of rodents living in complex environments date back to experiments by Peter Crowcroft that were conducted in large unheated enclosures containing hundreds of small objects and wooden housing structures (*Crowcroft, 1966*). A more contemporary solution to naturalistic captivity would be expanded versions of the visible burrow system (*Pobbe et al., 2010*). In this regard, biomedical researchers should collaborate with behavioral ecologists to develop captive environments that offer captive animals sufficient complexity and agency resembling that of their free-roaming counterparts. For species that naturally travel long distances, captivity may never adequately provide for their needs.

Another possible solution is to study smaller organisms that are less capable of realizing the limits of their captive environments. Though these organisms live in small-scale environments, natural conditions may be no less critical. For instance, zebrafish development is highly sensitive to the environmental complexity of their tanks (*Spence et al., 2011*), and mutants of the worm *C. elegans* that live twice the age of normal worms in the laboratory die earlier than their controls in natural soil (*Van Voorhies et al., 2005*). In these cases, biomedical researchers would benefit by collaborating with ecologists.

Alternatives to the use of caged animals also include 'human-on-a-chip' models, combinations of human cell types or 'organoids' that simulate cellular and molecular interactions and provide insight into many biological processes not otherwise obtainable (*Kilic et al., 2016*; *Dauth et al., 2017*). These alternatives are particularly attractive for many animal welfare advocates and they hold great promise. However, cell-based models do not always serve as adequate replacements for intact animals. For instance, we cannot learn about the complex relationships between brain activity and behavior, or between immune cell activity and disease resistance, if we only study cell cultures.

## Conclusions

If our goals to improve human health do justify our means of using research animals, then we should be very concerned that animals raised in captivity do not fare well in their natural environments and are hypersensitive to experimental manipulations. These inadvertent effects may impose huge costs for biomedical research as laboratory animals are often sensitive to drug treatments that are later found to be ineffective in human trials (*Pound et al., 2004*; *Hackam and Redelmeier, 2006*; *Tsilidis et al., 2013*; *Lee and Feng, 2005*). To improve our understanding of human health, we must attend to the wellbeing of our animal models.

Giving research animals agency within naturalistic environments poses difficult challenges for scientists, who have traditionally studied animals living inside cages, pens and corrals. This paradigm shift also poses difficulties for those animal welfare advocates who believe that compassionate care should minimize all temporary discomforts. Motivations for rewards can rebound after their access has been denied (*Panksepp et al., 2008*). The health and wellbeing of a research animal require risks and discomforts, just as for humans.

In a sense, research animals confined inside our conventional research environments resemble prisoners in the Allegory of the Cave, experiencing shadows of the real-life conditions in which they evolved (and what we attempt to model). By studying captive animals exclusively, we constrain our own abilities to understand the problems we aim to investigate. We also have a responsibility to interject the scientific process into a swelling public debate regarding the compassionate treatment of research animals. Incumbent upon us, we must identify the aspects of environmental complexity necessary for the wellbeing of our animal models, and their relevance to human health, lest we also risk attending to shadows on a cave wall.

**Garet P Lahvis** is in the Department of Behavioral Neuroscience, Oregon Health & Science University, Portland, United States

http://orcid.org/0000-0002-1986-8251

*Competing interests:* The author declares that no competing interests exist.

## Funding

No external funding was received for this work.

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
