## [Decision Letter]

Thank you for submitting your article "Unbridle Biomedical Research from the Laboratory Cage" to *eLife* for consideration as a Feature Article. Your article has been reviewed by 3 peer reviewers, and the evaluation has been overseen by a Reviewing Editor and Stuart King as the Senior Editor.

The reviewers have discussed the reviews with one another and I have drafted this decision to help you prepare a revised submission.

The following individuals involved in review of your submission have agreed to reveal their identity: Francoise Wemelsfelder and John Gluck.

Summary:

In this feature article, Garet Lahvis brings to our attention an important issue regarding biomedical research in animals. He presents a clear and startling array of evidence, carefully referenced, that animals kept in laboratory cages do not behave normally, and that free-ranging (wild) animals would be more representative models to work with. Lahvis proposes nothing less than a revolution of scientific thought, made possible by technological advances, where animal subjects are monitored and measured under conditions not of maximum control by humans, but of maximum feasible freedom and control by the animals themselves.

The article is timely, coming as it does in the context of attempts to understand the failure of many preclinical animal studies to translate into humans and the so-called reproducibility crisis. It also details what is missing in the experiences of captive lab animals that contributes to these failures and distorts attempts to improve captive animal welfare.

Essential revisions:

1) Please delete the first paragraph of the section entitled "Responding to a Responsive Environment", or rephrase to make the main points clearer. Additional comments from reviewer #1 "Yes, enrichment should be to scale with cognitive ability. Adding a tube to a mouse cage is not to scale with adding a tube to a monkey cage. Adding a tree to a monkey enclosure, for example, would be more akin to adding a tube to a mouse cage, allowing the animal to climb it, perch in it, or sit in its shade, each experience affording the monkey a unique 360-degree visual experience. Also, visual experience is perhaps not the best example for a mouse, given its poor eyesight; 'enrichment' affording unique thigmotaxic, olfactory or auditory experience would be more appropriate."

2) Please add the following reference to the section entitled "Not just Neurobiology": Martin et al., 2010. "Control" laboratory rodents are metabolically morbid: Why it matters. PNAS 107(13): 6127-6133.

3) In the section “Research (on) animal well-being”, please broaden the discussion to include other references. It may also be helpful to also (briefly) mention and discuss the recent revival of interest in, and acceptance of, the notions of animal sentience and positive welfare. See for example reviews by Boissy et al. (2007) and Yeates and Main (2008). It seems these widely accepted developments support the author's call for greater recognition of the nature of animal lives.

4) Telemetry and Solutions: This section is the culmination of the article's argument and should be expanded.

How could traditional biomedical research (e.g. modulating gene expression) be conducted in the field using these technologies?

How would using these technologies affect the wild animals involved (e.g. sensor implantation, experimental manipulations)?

Are alternatives to uses of animals in research (e.g. tissues on a chip) likely to be useful in overcoming the validity and reliability issues intrinsic to captive animal use?

At the very least, some references where some of these technologies are described should be provided so that readers can seek further information. This section should also include some discussion of experiments that would be difficult/impossible to do in wild animals due to ethical and biological concerns (e.g. deliberately exposing the animals to a disease to trial a new drug candidate).

5) Please clarify the sentence "A benefit of using standard laboratory mice and rats is that they have evolved to be commensal with humans and this commensalism has collapsed their home range". What is meant by standard, evolved (over what time frame?), and how does that relate to their home range being 280,000 fold smaller in a laboratory (is that before or after the collapse)?

6) Is the article arguing that caged animals should no longer be studied, or that they could be if we expanded our notions of good welfare to include negative experiences? Please clarify your position from the beginning of the article and make necessary changes in the manuscript to keep your position on this central issue clear. Note that many animal welfare scientists do suggest that good welfare includes exposing animals to manageable threats, i.e. challenges that they can overcome. This gives animals agency and an opportunity to exert control over the environment; this ties into issues described in the section “Research animal well-being”.

---

## [Author Response]

*Essential revisions:*

*1) Please delete the first paragraph of the section entitled "Responding to a Responsive Environment", or rephrase to make the main points clearer. Additional comments from reviewer #1 "Yes, enrichment should be to scale with cognitive ability. Adding a tube to a mouse cage is not to scale with adding a tube to a monkey cage. Adding a tree to a monkey enclosure, for example, would be more akin to adding a tube to a mouse cage, allowing the animal to climb it, perch in it, or sit in its shade, each experience affording the monkey a unique 360-degree visual experience. Also, visual experience is perhaps not the best example for a mouse, given its poor eyesight; 'enrichment' affording unique thigmotaxic, olfactory or auditory experience would be more appropriate."*

This paragraph has been deleted.

*2) Please add the following reference to the section entitled "Not just Neurobiology": Martin et al., 2010. "Control" laboratory rodents are metabolically morbid: Why it matters. PNAS 107(13): 6127-6133.*

This reference has been added. See subsection “Not just neurobiology”, first paragraph.

*3) In the section “Research (on) animal well-being”, please broaden the discussion to include other references. It may also be helpful to also (briefly) mention and discuss the recent revival of interest in, and acceptance of, the notions of animal sentience and positive welfare. See for example reviews by Boissy et al. (2007) and Yeates and Main (2008). It seems these widely accepted developments support the author's call for greater recognition of the nature of animal lives.*

I added the suggested references and expanded the discussion to include a more thorough description of the kinds of conditioning experiments that I believe give us the greatest insight to non-human subject experience. I also incorporated ideas from a few philosophers. Scientists tend to trust inferences about private subjective experiences based on human language over animal behavior, a point raised by Marilyn Dawkins. I added an argument by the philosopher Bertrand Russell, who reminds us why we should not trust language for making inferences regarding human subjective states. I also referenced a book of thoughtful essays on the ethics of captivity edited by Lori Gruen. See subsection “Research into animal wellbeing”, fourth paragraph.

4) Telemetry and Solutions: This section is the culmination of the article's argument and should be expanded.

How could traditional biomedical research (e.g. modulating gene expression) be conducted in the field using these technologies?

This section is now expanded to include more technical details. See subsection “What can we do”.

*How would using these technologies affect the wild animals involved (e.g. sensor implantation, experimental manipulations)?*

The essay now mentions that sensor implantation and experimental manipulations often require surgery, sedatives, and anesthetics. However, the harm of these procedures is diminishing See subsection “What can we do”.

*Are alternatives to uses of animals in research (e.g. tissues on a chip) likely to be useful in overcoming the validity and reliability issues intrinsic to captive animal use?*

The essay now addresses the value and limitations of tissues-on-a-chip. I was surprised to learn how advanced these in vitro technologies have become. Also mentioned is that in vitro studies have their limitations. We cannot, for example, learn how a particular change in the biology of a brain affects a certain behavior without an intact organism. See subsection “What can we do”.

*At the very least, some references where some of these technologies are described should be provided so that readers can seek further information. This section should also include some discussion of experiments that would be difficult/impossible to do in wild animals due to ethical and biological concerns (e.g. deliberately exposing the animals to a disease to trial a new drug candidate).*

Studies that entail manipulations causing illness to animal subjects would require high levels of monitoring and the capacity for rapid retrieval. These studies would be better suited for captivity within naturalistic environments. Captivity would also be necessary for gene-targeted animals that may pose a threat to the environment. See subsection “What can we do”.

*5) Please clarify the sentence "A benefit of using standard laboratory mice and rats is that they have evolved to be commensal with humans and this commensalism has collapsed their home range". What is meant by standard, evolved (over what time frame?), and how does that relate to their home range being 280,000 fold smaller in a laboratory (is that before or after the collapse)?*

To clarify the sentence regarding commensalism, the essay now more clearly states that mice and rats can live in close proximity to humans at higher population densities and that the comparison shown in the figure was based upon estimations of the natural home ranges of mice and rats. See subsection “What can we do”, sixth paragraph.

6) Is the article arguing that caged animals should no longer be studied, or that they could be if we expanded our notions of good welfare to include negative experiences? Please clarify your position from the beginning of the article and make necessary changes in the manuscript to keep your position on this central issue clear. Note that many animal welfare scientists do suggest that good welfare includes exposing animals to manageable threats, i.e. challenges that they can overcome. This gives animals agency and an opportunity to exert control over the environment; this ties into issues described in the section “Research animal well-being”.

My position is now clarified throughout. In consideration of their wellbeing and more relevant findings in biomedical research, animal models should be either studied in the wild or studied as captive subject under naturalistic conditions. See main text, first paragraph, subsection “Research into animal wellbeing”, last paragraph, and subsection “Conclusions, first paragraph).